# Amyloid Aggregates of Smooth-Muscle Titin Impair Cell Adhesion

**DOI:** 10.3390/ijms22094579

**Published:** 2021-04-27

**Authors:** Alexander G. Bobylev, Roman S. Fadeev, Liya G. Bobyleva, Margarita I. Kobyakova, Yuri M. Shlyapnikov, Daniil V. Popov, Ivan M. Vikhlyantsev

**Affiliations:** 1Institute of Theoretical and Experimental Biophysics, Russian Academy of Sciences, Pushchino, 142290 Moscow, Russia; fadeevrs@gmail.com (R.S.F.); liamar@rambler.ru (L.G.B.); ritaaaaa49@gmail.com (M.I.K.); yuri.shlyapnikov@gmail.com (Y.M.S.); 2State Research Center Institute of Biomedical Problems, Russian Academy of Sciences, 123007 Moscow, Russia; danil-popov@yandex.ru

**Keywords:** amyloid, amyloidosis, amyloid aggregation, cytotoxicity of amyloid fibrils, protein aggregation, smooth-muscle titin

## Abstract

Various amyloid aggregates, in particular, aggregates of amyloid β-proteins, demonstrate in vitro and in vivo cytotoxic effects associated with impairment of cell adhesion. We investigated the effect of amyloid aggregates of smooth-muscle titin on smooth-muscle-cell cultures. The aggregates were shown to impair cell adhesion, which was accompanied by disorganization of the actin cytoskeleton, formation of filopodia, lamellipodia, and stress fibers. Cells died after a 72-h contact with the amyloid aggregates. To understand the causes of impairment, we studied the effect of the microtopology of a titin-amyloid-aggregate-coated surface on fibroblast adhesion by atomic force microscopy. The calculated surface roughness values varied from 2.7 to 4.9 nm, which can be a cause of highly antiadhesive properties of this surface. As all amyloids have the similar structure and properties, it is quite likely that the antiadhesive effect is also intrinsic to amyloid aggregates of other proteins. These results are important for understanding the mechanisms of the negative effect of amyloids on cell adhesion.

## 1. Introduction

Misfolding of proteins and their subsequent aggregation is a well-known pathological process associated with such diseases as amyloidoses [1,2]. They include Alzheimer’s and Parkinson’s diseases, type II diabetes, prion diseases, as well as systemic amyloidoses [3,4,5,6]. The toxic effect of various intermediaries of amyloid aggregates, including mature fibrils, on cells and the organism is considered to be the main danger of amyloid structures [7,8,9,10,11,12]. 

Amyloid aggregates are divided into two classes: Amyloid fibrils of ordered structures and amorphous aggregates [13,14,15,16]. Amyloids have a number of specific characteristics, such as an ability to bind the dyes Congo Red and Thioflavin T (ThT); they are insoluble in most solvents and are stable to proteases [17,18]. The main property of amyloids, irrespective of the type of aggregates they form, is the presence of a quaternary cross-β structure in them [19,20]. In ordered amyloid aggregates, particular β-sheets are organized perpendicular to the fibril main axis of growth.

Numerous studies have shown amyloid aggregates of various proteins and peptides to impair the vital activities of cells and cause their death. Aggregates of many amyloid proteins exhibit cytotoxic properties [21,22,23,24]; however, the mechanisms of their toxic action are not yet finally clear. Research shows that mature amyloid fibrils [10], amorphous aggregates [11,25], and oligomers—products of the initial stage of amyloid aggregation [12,26,27,28]—are toxic. In this work, we investigated the action of amorphous amyloid aggregates of titin on cells.

Titin (connectin) is a multidomain muscle protein [29]. The molecular masses of titin isoforms in smooth and striated muscles are 500–3800 kDa [29,30,31]. In striated muscles, titin is part of the sarcomere and interacts both with myosin (thick) and actin (thin) filaments (Figure 1). The greatest part (up to 90%) of titin molecule are immunoglobulin (Ig)-like and fibronectin III (FnIII)-like repeats of a β-fold structure [32] (Figure 1B). Its molecules about 1 μm in length and 3–4 nm in diameter [33] (Figure 1C) span half of the sarcomere from the M-line to the Z-disk to form the third type of filaments called elastic. The localization of titin in smooth muscles remains unknown. Smooth-muscle titin (SMT) was found in a chicken gizzard smooth-muscle extract in 2002 [34]. Subsequent research has shown that smooth- and striated-muscle titin is the product of one gene whose alternative splicing leads to the formation of titin isoforms with molecular masses of 700–2000 kDa in smooth-muscle tissue [30]. Western blotting of human aortic smooth-muscle cells has also detected the presence of titin of a molecular mass of ~500 kDa, which is assumed to be either a titin isoform or a truncated fragment of this protein [30].

Previously, we have shown that 500-kDa SMT can form amorphous aggregates of a quaternary cross-β structure intrinsic to all amyloid aggregates [36,37]. We have also investigated the cytotoxicity of those aggregates [36]. It has been found that SMT amorphous aggregates have a toxic action on a culture of smooth-muscle cells [36].

In this work, we investigated the effect of amyloid aggregates of SMT (MW, 1500 kDa) on cell adhesion. Its impairment was found to be due to a change of the relief of the cell cultivation surface by SMT amyloid aggregates.

## 2. Results

### 2.1. SDS-PAGE and Mass Spectrometry of Purified Titin

Appendix A shows SDS-PAGE of purified chicken gizzard SMT. Preparations of purified titin contained two SMT isoforms of molecular weights of approximately 500 and 1500 kDa. Appendix A presents data of mass spectrometry analysis confirming the presence of titin in the samples.

### 2.2. Formation of SMT Aggregates and Confirmation of Their Amyloid Characteristics

Figure 2A shows amorphous aggregates of 1500-kDa SMT formed during 24 h at 4 °C in a solution of 0.15 M glycine-KOH, pH 7.2–7.4. During the binding of the ThT dye to the aggregates the fluorescence intensity of the dye was observed to increase as compared with that in the non-aggregated form of the protein (Figure 2B).

### 2.3. Cytotoxicity Study of SMT Aggregates

Figure 3 presents data on the cytotoxic effect of titin aggregates on a culture of smooth-muscle cells after 72 h. The viability of 50% cells (IC_50_) was observed at an SMT concentration of 70 μg/mL. There was no cell death in the presence of aggregates at lower concentrations.

### 2.4. Study of the Impact of SMT Aggregates on Actin Cytoskeleton

The effect of SMT aggregates on the actin cytoskeleton of rat aortic smooth-muscle cells (RAOSMCs) was studied by confocal microscopy. A disorganization of the cytoskeleton was found not until 48 h after the addition of SMT aggregates to RAOSMCs (Figure 4B). Morphological changes of cells were also observed, such as the occurrence of filopodia, stress fibers and lamellipodia. The results primarily indicate impaired adhesion of RAOSMCs in the presence of SMT aggregates. No actin cytoskeleton impairment or impaired cell adhesion were observed if non-amyloid actin filaments (F-actin) were added to the cell culture (Figure 4A).

### 2.5. Atomic Force Microscopy Study of the Morphology and Nanostructure of Surfaces Coated with Titin Amyloid Aggregates

To understand the causes of cell adhesion impairment, we studied the effect of the relief of the surface coated with titin amyloid aggregates on adhesion of fibroblasts. Figure 5 presents atomic force microscopy (AFM) images, which show the surface relief of cover glasses with applied SMT aggregates. Non-aggregated (molecular) titin adsorbed on the surface of mica at a high ionic strength (~0.6) was used as a control (Figure 5A). According to the literature data, SMT molecules look like filaments ~750–900 nm long. At the end of a filament, there is a globular head [36]. The AFM of control samples showed both separate SMT molecules morphologically similar to those of skeletal muscle titin (Figure 1C) and clusters of SMT molecules that may represent oligomers of the protein (Figure 5A).

Figure 5B,C presents 2-D and 3-D AFM images, which show the morphology and nanostructure of SMT aggregates. A significant roughness of the surfaces was observed. The roughness was determined by calculating the root mean square (RMS) roughness parameter. The calculated surface roughness values varied from 2.7 to 4.9 nm depending on the scan scale, which is much greater as compared with the cover-glass controls without adsorbed titin (RMS values less than 1 nm).

### 2.6. Study of the Effect of SMT-Aggregate-Coated Surface on Fibroblast Adhesion

Following the analysis of the relief of a surface coated with titin amyloid aggregates, we investigated the ability of fibroblasts to attach to and grow on this surface. For this, SMT amyloid aggregates were placed onto half of the cover-glass surface. Then human fibroblasts were seeded onto the entire cover-glass surface. The scheme of the experiment is shown in Figure 6.

We observed a significant decrease of the number of fibroblasts after 48 and 72 h of incubation relative to the control (Figure 7), accompanied with a partial or complete disorganization of the actin cytoskeleton. Fibroblasts growing on a surface coated with SMT aggregates featured morphological changes such as the presence of filopodia, as well as a large number of stress fibers and lamellipodia, which is indicative of an increased migration of cells. These results suggest impaired cell adhesion and, as a consequence, a decrease of the cell growth rate.

## 3. Discussion

The study revealed a cytotoxic action of amyloid aggregates of SMT (MW, 1500 kDa) on smooth-muscle cells. We have observed a similar effect earlier for aggregates formed from a 500-kDa smooth-muscle isoform of titin [36]. Thus, we demonstrate the cytotoxic action on cells of SMT amyloid aggregates not to depend on the molecular weight of the protein. The toxic effect was observed not until after 72 h of smooth-muscle cells’ incubation in a medium with SMT aggregates. With this in mind, we assumed that the cell death was not related to their direct toxic action causing necrosis or apoptosis. Still, the mechanism of this toxic action remained elusive. In the process of incubation in a medium with SMT amyloid aggregates, we found that after 24 h the morphology of cells changed, they acquired a ball-like shape. Confocal microscopy revealed a partial or complete disorganization of the actin cytoskeleton, as well as morphological changes commonly occurring in adhesion impairment, such as the emergence of filopodia, lamellipodia, and stress fibers. These morphological features are indicative of a possible impairment of cell adhesion [38,39,40].

In some cases, aggregates of amyloid β-proteins affect smooth-muscle cells [41]. It has been shown that Alzheimer’s disease is characterized by accumulation of amyloid in the media and adventitia of small and large arteries perfusing the central nervous system or by accumulation of amyloid β-protein deposits around perfusion capillaries of the cerebral cortex [42]. Also, accumulation of amyloid β-protein aggregates in arteries has been found to induce the death of vascular smooth-muscle cells (VSMC) and to lead to blood seepage through vessel walls due to vascular integrity impairment. This may result in cerebral ischemia and microhemorrhages as clinical features of cerebral amyloid angiopathies [43]. Using a smooth-muscle cell culture, it has been shown that the amyloid β-protein-induced negative action on VSMC is related to impaired adhesion of these cells to the extracellular matrix due to increased pericellular proteolysis of extracellular matrix components [44]. No other mechanisms of smooth-muscle cell death have been found [44]. In particular, activation of matrix metalloproteinase-2 (MMP-2) by aggregates of amyloid β-protein (1–40) has been excluded. It is noteworthy that those authors have shown that the negative effects and death of smooth-muscle cells are observed, as in our experiments, also after 72 h of incubation with amyloid β-protein (1–40).

There are literature data for oligomers of amyloid β-proteins on a neuroblastoma culture [45]. A reorganization of the actin cytoskeleton has been revealed, as well as the emergence of stress fibers and lamellipodia in cells under the action of amyloid β-protein oligomers. The cause of the revealed changes is assumed to be the activation of the Rho GTPase signaling pathway by amyloid β-protein.

Comparing the above-described data with those obtained for titin, it should be noted that amyloid β-protein studies deal with the effects on the culture of cells or monomer protein or its oligomers. However, studies of the in vivo pathological process concern depositions of this protein in capillaries. In this case, amyloid β-protein deposits are hardly separate oligomers or monomers; they apparently represent ensembles of aggregates with a cytotoxic action. Our studies revealed a cytotoxic action of titin aggregates. The aggregates caused an impairment of the actin cytoskeleton and adhesion of cells. Taking into account that titin aggregates are 5–10 μm and more in size, we assumed that the main cause of adhesion impairment and cell death could be the direct interaction of the aggregates of this protein with cell membrane proteins. Moreover, the aggregates could stick to the surface of plate wells and, thus, prevent cell migration and adhesion. The cells present on the surface of SMT aggregates could, probably, not adhere, the result of which were the observed morphological changes described above.

To verify this assumption, we investigated the ability of cells to attach to and grow on the surface of SMT amyloid aggregates. For this, a uniform layer of aggregated SMT was applied onto half of the surface of cover glasses. Human dermal fibroblasts (HDF) were chosen. Cells were seeded directly onto the cover-glass surface coated with titin amyloid aggregates and onto the free part of the surface, which served as a control (Figure 6). Fibroblasts are the most numerous cells in various tissues of the organism. They play an important part in tissue angiogenesis and regeneration [46,47,48]. Fibroblasts secrete various growth factors and angiogenic factors, e.g., fibronectin, transforming growth factor-beta1 (TGF-β1), basic fibroblast growth factor (bFGF), collagens I and II, and connective-tissue growth factor. This feature makes fibroblasts key players in the control of extracellular medium, as well as in the regulation of the behavior of adjacent cells and their reaction to the surrounding medium.

In our experiment, after 48 and 72 h of HDF growth on the surface of a cover glass coated with SMT amyloid aggregates, we registered a decrease in the number of cells relative to the control (Figure 7). As in the case of the smooth-muscle culture, in fibroblasts we observed a disorganization of the actin cytoskeleton and morphological changes such as stress fibers, lamellipodia, and filopodia. No similar changes were found in fibroblasts growing on the control surface of the cover glasses (Figure 7A). Thus, we confirmed our assumption that the properties of the surface represented by aggregated protein may affect cell adhesion. The surface coated with SMT amyloid aggregates is an anti-adhesive surface, and the protein aggregates themselves possess anti-adhesive properties.

Understanding the process of cell attachment to any surface is related to understanding the sequence of physico-chemical reactions between cells and the surface [49]. The main presumed mechanisms of the anti-adhesive properties are surface energy, electrostatic interaction, steric repulsion, hydration, and relief topography of the surface [50,51,52,53,54]. Cells interact with the surface via a layer of proteins, which they themselves secrete, e.g., cell fibronectin [55]. Their adhesion on the surfaces of these proteins is provided for by cell adhesion receptors, mainly integrins [38,56,57].

Cells interact with the adhesion surface in different ways depending on the distance between the cell membrane and the surface. At a distance of greater than several micrometers, there is no interaction, and cells are of typical spherical shape. A distance of ~1 μm initiates a surface recognition process mediated by weak non-specific interactions between the pericellular envelope and the adhesion surface [58]. This stage takes place within tenths of fractions of a second. Weak and cooperative interactions contribute to stronger contacts. At this stage the distance between the cell membrane and the surface reaches hundreds of nanometers, and interactions are mediated by cell membrane proteins, integrins, which recognize certain molecular motifs on the adhesion surface. This stage occurs within several seconds. Depending on the density of adhesion motifs, their distribution and mechanical properties, the cell may begin to build larger and more stable molecular complexes to improve membrane attachment. After that the distance from the adhesion surface to the cell membrane reduces to tens of nanometers. This stage is called intermediate attachment or membrane adhesion; it includes the assembly of the cell cytoskeleton and occurs with a time interval of tens of seconds. Finally, the late adhesion or the cell spreading state is initiated by the establishment of mature adhesion molecular clusters [59]. On the whole, cell adhesion is a complex process, affected by numerous other factors apart from the type of cells themselves [55]. They include physico-chemical parameters of the surface such as free energy, electrostatic interactions, steric repulsion, hydration, and topography [60,61,62,63,64].

The topography of polymer surface on the micro- and nano-level affects the interaction of cells or proteins with the polymer. In particular, studies are known which determine the effect of surface topology on the morphology, adhesion, migration, and differentiation of stem cells [54,65,66]. The higher the surface roughness coefficient is, to a greater extent the cell adhesion is impaired, and the lower the affinity of cells to the surface is [67].

Surface irregularities formed by SMT amyloid aggregates were investigated by atomic force microscopy. The obtained RMS roughness values were from 2.7 up to 4.9 nm, which is indicative of a high roughness. These values exceed those at which the cell adhesion has been observed to be significantly impaired [67]. These data and our results suggest that the high degree of surface roughness formed by SMT amyloid aggregates is the cause of fibroblasts’ impaired adhesion. The process is schematically shown in Figure 8.

There is no answer yet to the question about the exact mechanism of impaired adhesion of smooth-muscle cells and fibroblasts to the surface of SMT amyloid aggregates. Various other studies on the effect of microtopology on the anti-adhesive properties give no answer to this question, either [59,68,69,70]. However, topographic effects should not be considered as in vitro artefacts. Deposits of salt crystals in bones may serve as examples of rough surfaces in vivo.

Discussing the obtained results, it can be assumed that the local formation of amyloid aggregates in vivo may result in changes in the topography of the substrate followed by the adhesion impairment leading to cell death. It is quite probable that such a process occurs at the accumulation of amyloid β-protein deposits in vessels [41,44]. It should be noted that in the organism cells are in a 3-D microenvironment. Under these conditions, the inhibitory effect of amyloid structures on adhesion/spreading and, correspondingly, migration of cells can presumably be more pronounced than in a 2-D microenvironment—under 3-D conditions the impaired adhesion signal comes from the entire cell surface, not only from its basal part, as in 2-D. We cannot assert, however, that amyloid aggregates formed in vitro will correspond by their structure and functions to in vivo aggregates. Other mechanisms affecting cell adhesion cannot be ruled out, either. Nevertheless, the presented route of cell death in the presence of amyloid aggregates that impair cell adhesion expands our views on the effect of amyloids on the organism.

## 4. Materials and Methods

### 4.1. Purification of Chicken Gizzard SMT

Smooth-muscle titin was extracted from fresh chicken gizzard as described in [34] with our modifications [33]. In particular, fresh chicken-gizzard smooth muscles were diced and homogenized for 10 s by a Waring blender in buffer A (2 mM MgCl_2_, 1 mM EDTA, 1 mM EGTA, 0.5 mM DTT, 0.2 mM PMSF and 0.1% cocktail protease inhibitors, 10 mM imidazole, pH 7.0, containing 50 mM KCl). Gizzard myofibrils were pelleted by centrifugation (5000× *g* for 10 min, 4 °C), washed three times with buffer A and resuspended in extraction buffer (2 mM MgCl_2_, 1 mM EDTA, 1 mM EGTA, 0.5 mM DTT, 0.2 mM PMSF and 0.1% cocktail protease inhibitors (Sigma-Aldrich, St. Louis, MO, USA), 10 mM imidazole, pH 7.0, containing 0.6 M KCl and 4 mM ATP) for 60 min (final ionic strength, ~0.4). The extract was clarified for 30 min at 15,000× *g*, and the supernatant was diluted twofold (final ionic strength, ~0.2) with cooled double distilled water containing 0.1 mM DTT and 0.1 mM NaN_3_ to precipitate actomyosin. After 1 h, the supernatant was clarified for 60 min at 20,000× *g*. To precipitate SMT, the supernatant was diluted fourfold (final ionic strength, ~0.05) with cooled double distilled water containing 0.1 mM DTT and 0.1 mM NaN_3_. After 40–60 min, the residue containing predominantly SMT was collected by centrifugation for 30 min at 15,000× *g*. The residue was dissolved in a minimal volume of the buffer containing 0.6 M KCl, 30 mM KH_2_PO_4_, 1 mM DTT, 0.1 M NaN_3_, pH 7.0 and clarified for 60 min at 20,000× *g*. SMT was purified by gel filtration on a Sepharose-CL2B (Sigma-Aldrich, St. Louis, MO, USA) column equilibrated in the same buffer. 

### 4.2. SDS-PAGE and Mass Spectrometry Analysis of Titin

The presence of titin in the sample was confirmed by SDS-PAGE (Appendix A) and mass spectrometry analysis (Appendix A). SDS-PAGE of titin was performed as in [71] with our modification. In particular, our experiments used a separating gel containing 6.5–7% polyacrylamide prepared as described in [71]. For shotgun mass spectrometry analysis the sample was solubilized into a buffer (4% sodium dodecyl sulfate in 0.1 M Tris-HCl pH 7.6, 0.1 M dithiothreitol) and incubated for 5 min at 95 °C as described [72]. The samples were sonicated (4 × 30 s at 20 W; ME220, Covaris, Woburn, MA, USA), centrifuged (5 min, 16,000× *g*), and the supernatant was collected. The YM-30 filter (Millipore, Ireland) was used for alkylation and trypsinolysis (14 h, 2 μg of trypsin (Tripsin Gold, Promega, Madison, WI, USA)) according to the FASP method [73]. Peptides were desalted using C18 microcolumns and subjected to HPLC-MS/MS analysis using the HPLC Ultimate 3000 RSLCnano system (Thermo Scientific, Waltham, MA, USA) as described [72]. For a list of detected titin peptides, see Appendix A.

### 4.3. Isolation and Purification of Actin and Determination of the Concentration of Isolated Proteins

Actin was isolated as described in [74]. The concentrations of proteins used in the work were determined spectrophotometrically by a SPECORD UV VIS spectrophotometer (Carl Zeiss, Jena, Germany) using the extinction coefficient (*E*_280_^1 mg/mL^) of 1.08 for actin [75] and 1.37 for titin [76]. SDS-PAGE of purified actin was performed as in [77] (Appendix A).

### 4.4. Formation of SMT Aggregates and F-Actin Fibrils

Purified SMT in column buffer (0.6 M KCl, 30 mM KH_2_PO_4_, 1 mM DTT, 0.1 mM NaN_3_, pH 7.0) was used to form aggregates. SMT aggregates (concentration, 0.2–0.4 mg/mL) were formed by dialysis in Sigma-Aldrich cellulose membrane tubing (Cat. No, D9777; size, 25 × 16 mm) for 24 h at 4 °C against solutions containing 0.15 M glycine-KOH, pH 7.2–7.4. F-actin fibrils were formed by dialysis in a solution containing 0.1 M KCl, 1 mM DTT, 0.1 mM NaN_3_, pH 7.0.

### 4.5. Fluorescence Analysis

The amyloid nature of SMT aggregates was assessed by the intensity of ThT (Sigma-Aldrich, St. Louis, MO, USA) fluorescence (1 ThT: 5 SMT (*w*/*w*)). The fluorescence was measured at λ_ex_ = 440 nm and λ_em_ = 488 nm using a Cary Eclipse spectrophotometer (Varian, Palo Alto, CA, USA).

### 4.6. Electron Microscopy

A drop of a suspension of the protein at a concentration of 0.1 mg/mL was applied to a carbon-coated collodion film (2% collodion solution in amyl acetate (Sigma-Aldrich, St. Louis, MO, USA)) on a copper grid (Sigma-Aldrich, St. Louis, MO, USA) and negatively stained with a 2% aqueous uranyl acetate (SPI-Chem., West Chester, PA, USA) solution. Samples were examined on a JEM-100B electron microscope (JEOL Ltd., Tokyo, Japan) at an accelerating voltage of 120 kV.

### 4.7. Cytotoxicity Assay

To study the cytotoxicity of SMT aggregates, the protein was lyophilized using a FreeZone 1L freeze dryer (Labconco, Kansas, MO, USA). Trehalose (1%) was used as a stabilizing agent. The quality and integrity of SMT with respect to its possible degradation after the freeze drying were tested by gel electrophoresis in 7% polyacrylamide [71]. The cytotoxic action of SMT aggregates was assessed on rat aortic smooth-muscle cells by staining cells with Crystal Violet [78]. For this, cells were seeded into 96-well plates and cultured in a DMEM/F12 medium (Sigma-Aldrich, St. Louis, MO, USA) with addition of 10% FBS (Gibco, Gaithersburg, MD, USA), 40 μg/mL gentamicin sulfate (Sigma-Aldrich, St. Louis, MO, USA) at 37 °C and in a 5% CО_2_ atmosphere to a confluent state. Then cells were kept for 2 h in a DMEM/F12 medium without serum, and SMT in a molecular and aggregated form was added, or F-actin as a control. The cytotoxic effect of SMT aggregates was assessed by the ratio of the difference between the optical density in the experiment and the background to the difference between the optical density in the control and the background after 72 h of incubation. The measurement was carried out using an Infinite F200 Microplate Reader (Tecan, Grödig, Austria).

### 4.8. Confocal Microscopy

RAOSMCs were seeded onto cover glasses placed into 6-well plates. In 24 h after the seeding, the DMEM/F12 medium (Sigma-Aldrich, St. Louis, MO, USA) with addition of 10% FBS (Gibco, Gaithersburg, MD, USA) and 40 μg/mL gentamicin sulfate (Sigma-Aldrich, St. Louis, MO, USA) was replaced with a medium without the serum. After holding for 2 h in the medium without the serum, aggregated SMT was added, or F-actin (actin filaments) as a control. After a 48-h incubation at 37 °C and in a 5% CО_2_ atmosphere, cells were washed three times with cold PBS and fixed with 4% paraformaldehyde for 2 h at room temperature. Then cells were permeabilized with a 0.1% solution of saponin (Sigma-Aldrich, St. Louis, MO, USA). Fibrillar actin of the RAOSMC cytoskeleton was stained in a medium of the following composition: PBS, 1% BSA, 0.25 nmol Phalloidin-Atto 488 (Sigma-Aldrich, St. Louis, MO, USA) for 25 min in the dark at room temperature. Cell nuclei were stained with Hoechst 33342 (1 μg/mL) (Sigma-Aldrich, St. Louis, MO, USA). Then the cells were washed sixfold with a PBS solution. Further on, samples were dried in air and fixed using a Bio Mount synthetic mounting medium (Bio-Optica, Milano s.p.a., Milan, Italy). The samples were investigated by a TCS SP5 confocal microscope (Leica, Wetzlar, Germany).

### 4.9. Study of the Effect Of Titin Amyloid Aggregates on Adhesion of Fibroblasts

To study the effect of titin aggregates on adhesion of cells, the protein was lyophilized using a FreeZone 1L freeze dryer (Labconco, Kansas, MO, USA) with the view to subsequently produce higher-concentration protein preparations. As a stabilizing agent, 1% trehalose was used. The quality and integrity of SMT with respect to its possible degradation after the freeze drying were tested by gel electrophoresis in 7% polyacrylamide [71]. The lyophilized protein was dissolved in deionized water and dialyzed against a low ionic strength buffer (0.15 M glycine-KOH, pH 7.2–7.4) with the view to form amyloid aggregates. After that, aggregates of the protein (concentration, ~10 mg/mL) were applied onto sterile glasses to form protein films. Further on, after the protein films dried up, the glasses were washed with the culture medium.

The cell adhesion test was performed using human dermal fibroblasts on titin amyloid aggregate-coated glass substrates.

Human dermal fibroblasts were freshly harvested and resuspended in the cell culture media at a concentration of 5 × 10^4^ cells/mL. The modified glass substrates (10 × 10 mm^2^) were placed in 9-well culture plates and sterilized by UV irradiation for 30 min. Then the glasses were coated with a DMEM/F12 medium. The HDF were cultured in a DMEM/F12 medium (Sigma-Aldrich, St. Louis, MO, USA) with addition of 10% FBS (Gibco, Gaithersburg, MD, USA) and 40 μg/mL gentamicin sulfate (Sigma-Aldrich, St. Louis, MO, USA) at 37 °C under conditions of a 5% CО_2_ atmosphere for 48 or 72 h. Further on, after 48 or 72 h of incubation at 37 °C in a 5% CО_2_ atmosphere, cells were washed three times with cold PBS and fixed with 4% paraformaldehyde for 2 h at room temperature. Then cells were permeabilized with a 0.1% solution of saponin (Sigma-Aldrich, St. Louis, MO, USA). Fibrillar actin of the HDF cytoskeleton was stained in a medium of the following composition: PBS, 1% BSA, 0.25 nmol Phalloidin-Atto 488 (Sigma-Aldrich, St. Louis, MO, USA) for 25 min in the dark at room temperature. Cell nuclei were stained with Hoechst 33342 (1 μg/mL) (Sigma-Aldrich, St. Louis, MO, USA). Then cells were washed six times with a PBS solution. Further on, samples were dried in air and fixed using a Bio Mount synthetic mounting medium (Bio-Optica, Milano s.p.a., Milan, Italy). The samples were investigated by a TCS SP5 confocal microscope (Leica, Wetzlar, Germany).

### 4.10. Atomic Force Microscopy

Titin samples were applied onto a freshly cleaved mica or glass for 5 min, washed with deionized water (milli-Q) and dried in a stream of nitrogen. AFM images were obtained using a SmartSPM^TM^ 1000 atomic force microscope (AIST-NT, Moscow, Russia) with fpN11 cantilevers (SRIPP, Zelenograd, Russia; tip curvature radius, 10–25 nm). The tapping mode with a resonance frequency of 120–190 kHz was used in scanning experiments. Gwyddion software (Czech Metrology Institute, Brno, Czech Republic) was used for RMS calculations.

### 4.11. Statistics

The data are presented as means (*M*) and their standard errors (*m*). The statistical significance was determined by the ANOVA technique followed by the Newman–Keuls test and Student’s *t*-test.

## 5. Conclusions

We showed that amyloid aggregates of smooth-muscle titin caused the death of smooth-muscle cells. Taking into account a large molecular mass of titin (oligomers can have molecular masses of up to 10 MDa), we assumed that the negative action of its aggregates on cells was likely not due to the direct cytotoxic action. This was also indicated by the long incubation time in the presence of amyloid aggregates (72 h), after which cells died. The obtained confocal microscopy data, such as the reorganization of the actin cytoskeleton, occurrence of filopodia, lamellipodia, and stress fibers indicated the impaired adhesion of smooth-muscle cells. Presumably, during migration smooth-muscle cells creep onto protein’s amyloid aggregates stuck to the cover-glass support. This could impair adhesion. Additional studies that revealed an impairment of the ability of fibroblasts to grow on the surface of glass supports coated with SMT amyloid aggregates confirmed this assumption. Thus, we revealed an impairment of fibroblasts to a surface coated with amyloid aggregates of titin. AFM studies of the surface of amyloid aggregates found a rather large roughness. The calculated RMS values of roughness were 2.7–4.9 nm, which is characteristic of substrates with high anti-adhesive properties. Thus, we consider the death of cells in the presence of titin amyloid aggregates to be determined by impaired adhesion and to be related to, most probably, the microtopography of the surface. Titin in our studies is a model object of amyloid aggregates. Considering that the structure of amyloid aggregates of most proteins is similar, it can be assumed that the accumulation of amyloid aggregates in the organism may lead to a disturbance of adhesion of adjacent cells. We hope that this research opens prospects for further studies of pathologies associated with amyloidoses.

## Figures and Tables

**Figure 1 ijms-22-04579-f001:**
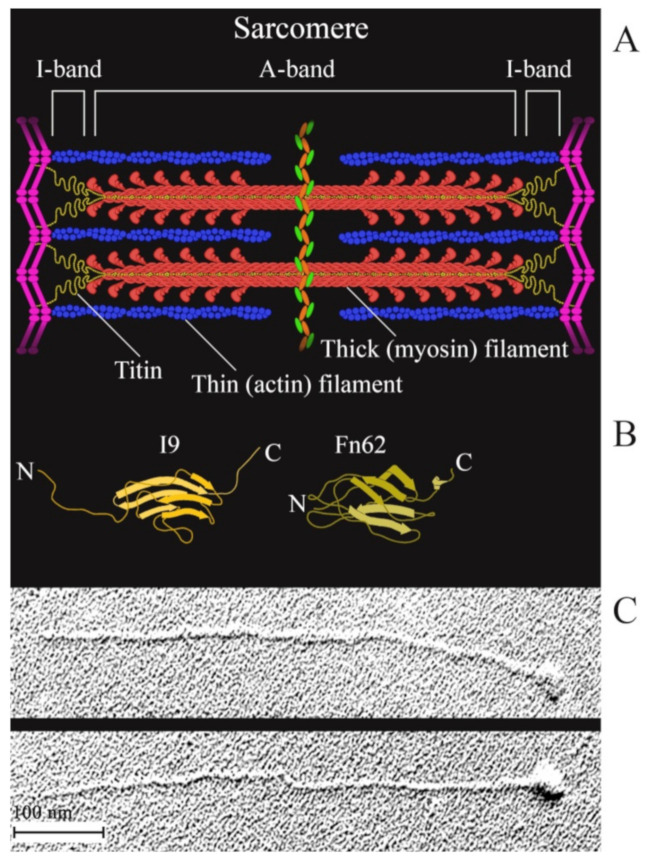
(**A**) a schematic structure of a sarcomere; (**B**) a schematic diagram of the structure of an Ig-like (I9) domain located in the I-band of the sarcomere and of an FnIII (Fn62) domain located in the A-band of the sarcomere. The domains were drawn based on the PDB file [35]; (**C**) electron micrographs of single straightened rabbit skeletal-muscle titin molecules produced as in [33].

**Figure 2 ijms-22-04579-f002:**
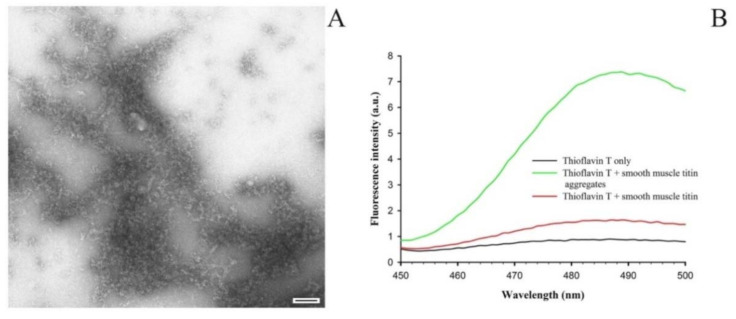
(**A**) An electron micrograph of SMT amorphous aggregates formed in a solution containing 0.15 M glycine-KOH, pH 7.2–7.4 (24 h; 4 °C); scale bar, 100 nm. (**B**) Binding of SMT aggregates to the ThT dye.

**Figure 3 ijms-22-04579-f003:**
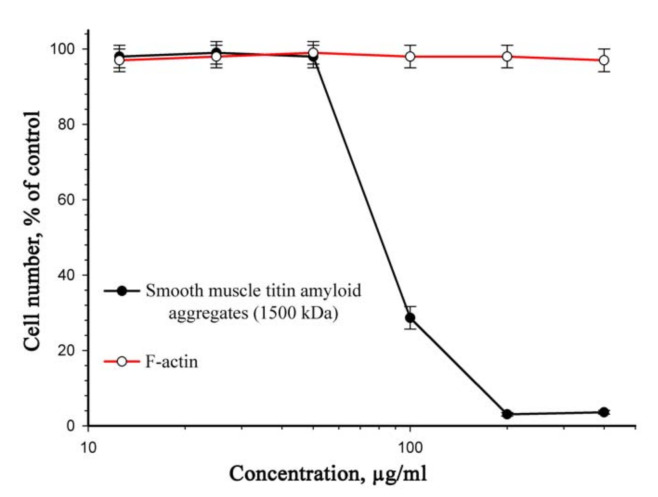
Cytotoxic effect of SMT amyloid aggregates on a rat aortic smooth-muscle cell culture (incubation, 72 h). F-actin was used as a control. Values are given as *M* ± *m*, *n* = 5 (see Section 4.11). SDS-PAGE of purified actin is presented in Appendix A.

**Figure 4 ijms-22-04579-f004:**
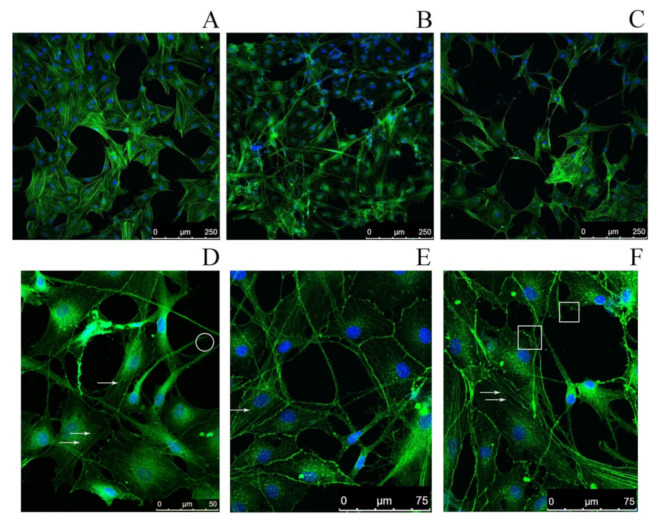
Confocal microscopy of RAOSMCs stained with Phalloidin-Atto 488 for visualization of the actin cytoskeleton (green) and with Hoechst 33342 (blue) to visualize cell nuclei. (**A**) Smooth-muscle cells upon addition of actin filaments (control); (**B**) smooth-muscle cells upon addition of 1500-kDa SMT amyloid aggregates, incubation time 48 h; (**C**) smooth-muscle cells in the presence of 1500-kDa SMT amyloid aggregates, incubation time 72 h. (**D**–**F**) High-magnification images of smooth-muscle cells upon addition of 1500-kDa SMT amyloid aggregates, incubation time 72 h. Arrows show stress fibers; in the circle, filopodia; in the squares, lamellipodia.

**Figure 5 ijms-22-04579-f005:**
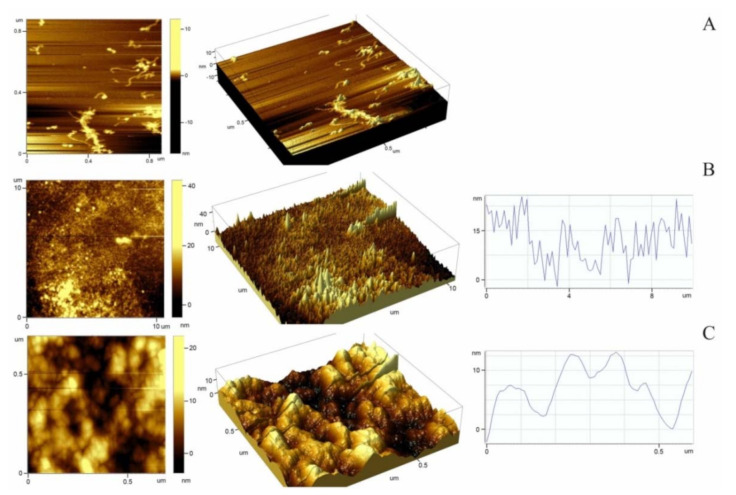
Atomic force microscopy of SMT molecule and aggregates. (**A**) 2-D and 3-D images of smooth-muscle titin molecules in a solution containing 0.6 M KCl, 30 mM KH_2_PO_4_, 1 mM DTT, 0.1 M NaN_3_, pH 7.0; (**B**,**C**) 2-D and 3-D images and section plots of 10 μm^2^ and 0.78 μm^2^ scans for titin-amyloid-aggregate-coated surfaces in air. RMS roughness: Scale 10 μm, 4.9 nm (**B**); scale 0.5 μm, 2.7 nm (**C**).

**Figure 6 ijms-22-04579-f006:**
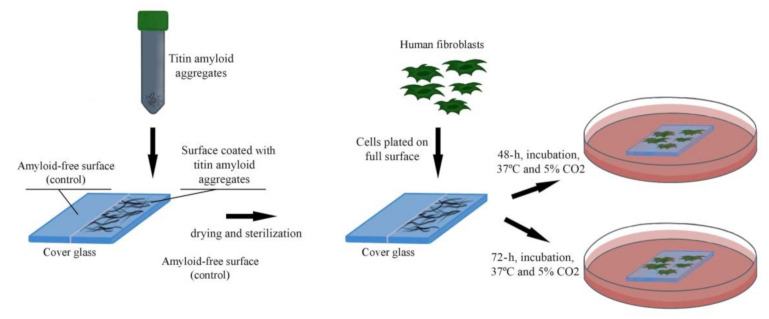
Growth of fibroblasts on a surface coated with SMT amyloid aggregates.

**Figure 7 ijms-22-04579-f007:**
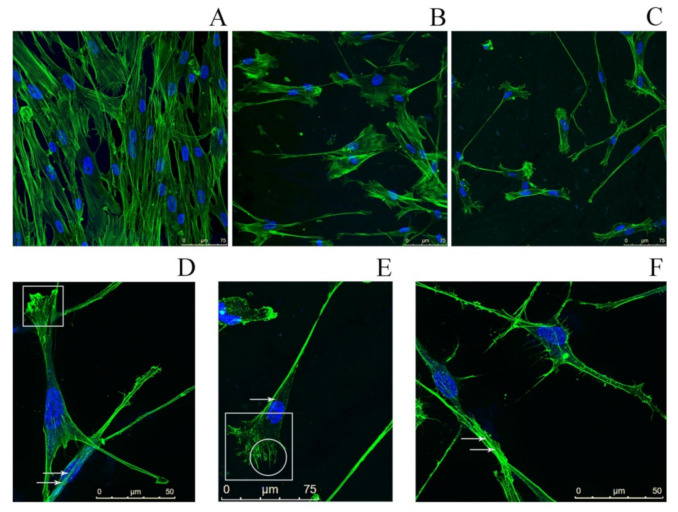
Fluorescence microscopy images of human dermal fibroblasts. (**A**) Fibroblasts (control) growing on a surface free of titin amyloid aggregates; staining with Phalloidin-Atto 488 (green) to visualize the actin cytoskeleton and with Hoechst 33342 (blue) to visualize cell nuclei. (**B**,**C**) Fibroblasts growing on a cover-glass surface with applied SMT aggregates (**B**) 48 h cultivation; (**C**) 72 h cultivation). (**D**–**F**) High-magnification images of fibroblasts growing on a cover-glass surface with applied SMT aggregates, incubation time 72 h. Arrows show stress fibers; in the circle, filopodia; in the squares, lamellipodia.

**Figure 8 ijms-22-04579-f008:**
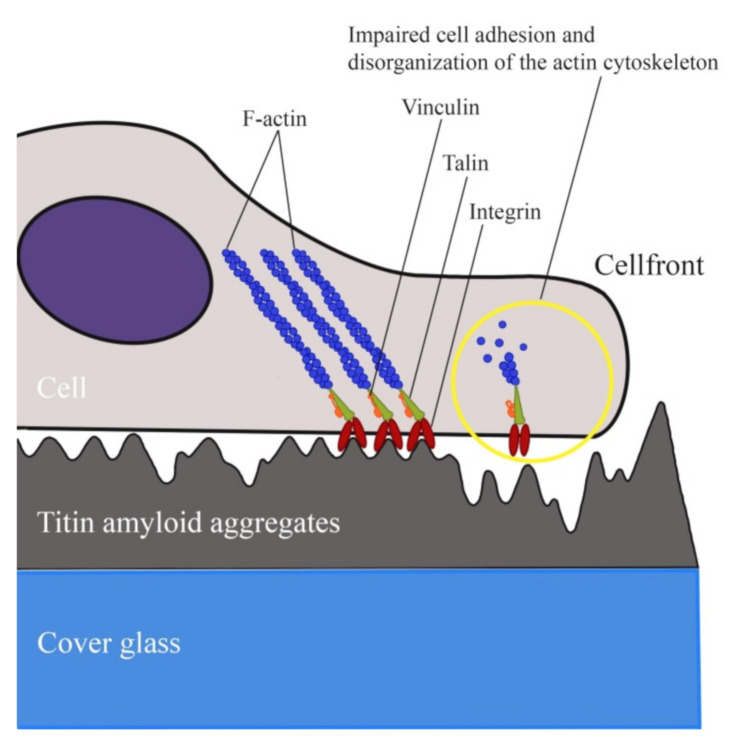
Schematic of a cell occurring on a surface coated with SMT amyloid aggregates. A possible topology-associated cell adhesion mechanism is shown.

## Data Availability

All data generated or analyzed in the course of this research (including files of additional information) were incorporated into the article and Appendix A.

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
