# Peer review of "Amyloid Aggregates of Smooth-Muscle Titin Impair Cell Adhesion"

_ijms, 2021, doi:10.3390/ijms22094579_

Round 1

Reviewer 1 Report

The manuscript by Bobylev et al. is written relatively well and present sound scientific analyses and findings. However, it is not publishable as presented. I have some comments below and tracked changes suggested in the PDF file to help the authors improve their manuscript.

Comments:

  1. The corresponding author should provide an official email address instead of a Gmail address.
  2. As I recall, the β-sheets are organised perpendicular (not in parallel to) the fibre axis. Please correct this factual mistake.
  3. It seems references 17 and 18 are not the original manuscripts describing the uniform characteristics of the amyloid proteins. Please refer to the primary sources.
  4. Despite being the commonly used and commonly accepted “misnomers”, the terms “Parkinson’s ‎Disease” (PD) and “Alzheimer’s Disease” (AD) are logically incorrect. The diseases were ‎discovered/described by Charles Parkinson or Alois Alzheimer, respectively; the diseases were not “their ‎own” diseases. Because of the eponymous convention, using the possessive form ‎‎(apostrophe plus ‘s’ or the genitive ‘s’) is wrong but has been perpetuated in the English ‎Scientific literature by our great peers. The Australian Manual of Scientific Style and The ‎Chicago Manual of Style advise against the use of the possessive form. I suggest taking their ‎editorial advice and using ‘Parkinson Disease’ ‎or ‘Alzheimer Disease’ instead.‎
  5. In Figure 1, please specify the I-zone and A-zone of the sarcomere. Are these the same as I band, and A band commonly used in describing the sarcomere in the literature and in textbooks? If so, please use the common terms.
  6. In the caption of Figure 3, ‘smaller times’ is written, but I think the authors mean ‘lower concentrations’. Please correct.
  7. Some abbreviations have been used in the text but not listed at the end of the manuscript, for example, RMS, FASP, and FBS.
  8. Standard error of the mean is designated as SEM. Revise.
  9. In Figure 4, the end point of observations was presented at 48 h. Please provide a time-course to show when the effects first appeared. Showing time points earlier than 48 h and 72 h will be informative. The confocal images are beautiful; however, provide high-magnification images to enable labelling (by arrows or arrowheads) of stress fibres, lamellipodia, and filopodia. This also relates to Figure 7. A few high-magnification images will add to value to the described observations.
  10. Change ‘irrigating’ to ‘perfusing’. More suggestions about the English usage in the Tracked PDF document.
  11. In the methods section, change all the mentions of the g-force to ‘value followed by a multiplication sign and then the italicized “g”’.
  12. When referring to a previously published method, do not use ‘method of [70]’. Use ‘as described previously {70], for example.
  13. Change the PubMed links in the methods section to properly formatted references with numbers so these are listed in the reference list.
  14. The findings presented in the supplementary files have not been described or referred to in the main text. Mention the findings and the significance of those experiments in the main text.
  15. Present a time course of the ThT assay to show the development of β-sheet structures overtime (likely in a supplementary figure).
  16. Make a distinction between ‘and’ and ‘or’ when two experimental conditions have been tested separately. This calls for ‘or’ instead of ‘and’. See tracked PDF.
  17. Define mQ.
  18. Discuss the limitations of the experimental design in your study. An obvious limitation is use of coverslips for growing the cells on two dimensions. Cells do not normally grow or function in two dimensions, considering the complex architecture of an organismal tissue. Discuss such limitations and their implications for the interpretation of the results. Discuss how the findings may be expanded or better described using a spheroid model of cell culture. This is important considering the new cell-culture models now used in the stem-cell field and cancer research, for example.
  19. Discuss how do the RMS roughness values of 2.7–4.9 nm compare with an average cell dimensions grown in 2-D or 3-D culture models? Please discuss and provide information to make the context clear.
  20. Discuss the implications of the surface roughness in thrombotic (fibrin) and atherosclerotic plaques? How do these compare with the model you used?
  21. See tracked changes for examples of English-language changes in the text. I suggest seeking assistance from an academic English editor before resubmission.

Author Response

Reviewer 1

  1. The corresponding author should provide an official email address instead of a Gmail address.

Answer:

The corresponding author has an official email address [email protected]. However, due to massive spam issues, that address has not been in use since about 5 years ago. The first author, Alexander Bobylev, has not registered an official email address for the same reason. We condider gmail addresses the most reliable and convenient. We have ORCID IDs, which are linked to the Institute:

Ivan Vikhlyantsev 0000-0001-6063-6789,

Alexander Bobylev 0000-0002-9477-1214

We would not object our ORCID IDs to be given in the article.  

Reviewer 1

  1. As I recall, the β-sheets are organised perpendicular (not in parallel to) the fibre axis. Please correct this factual mistake.

Answer:

Apologies for the error – indeed, β-sheets are organised perpendicular to the fibre axis. Error corrected, thanks!

Reviewer 1

  1. It seems references 17 and 18 are not the original manuscripts describing the uniform characteristics of the amyloid proteins. Please refer to the primary sources.

Answer:

References 17 and 18 were replaced by earlier works (reviews), which, in our mind, are original.  

Reviewer 1

  1. Despite being the commonly used and commonly accepted “misnomers”, the terms “Parkinson’s ‎Disease” (PD) and “Alzheimer’s Disease” (AD) are logically incorrect. The diseases were ‎discovered/described by Charles Parkinson or Alois Alzheimer, respectively; the diseases were not “their ‎own” diseases. Because of the eponymous convention, using the possessive form ‎‎(apostrophe plus ‘s’ or the genitive ‘s’) is wrong but has been perpetuated in the English ‎Scientific literature by our great peers. The Australian Manual of Scientific Style and The ‎Chicago Manual of Style advise against the use of the possessive form. I suggest taking their ‎editorial advice and using ‘Parkinson Disease’ ‎or ‘Alzheimer Disease’ instead.‎

Answer:

We understand – and agree with – Reviewer’s logic; corrections were made. Many thanks!

Reviewer 1

  1. In Figure 1, please specify the I-zone and A-zone of the sarcomere. Are these the same as I band, and A band commonly used in describing the sarcomere in the literature and in textbooks? If so, please use the common terms.

Answer:

Yes, these are I-band and A-band. Corrected.

Reviewer 1

  1. In the caption of Figure 3, ‘smaller times’ is written, but I think the authors mean ‘lower concentrations’. Please correct.

Answer:

Thanks! Correction made.

Reviewer 1

  1. Some abbreviations have been used in the text but not listed at the end of the manuscript, for example, RMS, FASP, and FBS.

Answer:

The abbreviations RMS, FASP, FBS and HDF were added to the list.

Reviewer 1

  1. Standard error of the mean is designated as SEM. Revise.

Answer:

Changed to M ± m in the legend to Fig. 3.

Reviewer 1

  1. In Figure 4, the end point of observations was presented at 48 h. Please provide a time-course to show when the effects first appeared. Showing time points earlier than 48 h and 72 h will be informative. The confocal images are beautiful; however, provide high-magnification images to enable labelling (by arrows or arrowheads) of stress fibres, lamellipodia, and filopodia. This also relates to Figure 7. A few high-magnification images will add to value to the described observations.

Answer:

In Fig. 4, the end point of observations is 72 h (С) and 48 h (B). Respective changes and corrections were made in the figure legend, as well as in the body of the article. The end point of 24 h is not given, as there are no distinctions from the control after 24 h incubation.

Similar results concerning the changes being observed only after 48 – not 24 – hours of the experiment are given for Fig. 7 (B, 48 h cultivation; C, 72 h cultivation).

Following Reviewer’s advice, we used greater-magnification confocal images and entered designations of stress fibres, lamellipodia, and filopodia. Thanks!

Reviewer 1

  1. Change ‘irrigating’ to ‘perfusing’. More suggestions about the English usage in the Tracked PDF document.

Answer:

This change was made, but we received no tracked PDF document.

Reviewer 1

  1. In the methods section, change all the mentions of the g-force to ‘value followed by a multiplication sign and then the italicized “g”’.

Answer:

Corresponding changes made.

Reviewer 1

  1. When referring to a previously published method, do not use ‘method of [70]’. Use ‘as described previously {70], for example.

Answer:

Several respective changes made.

Reviewer 1

  1. Change the PubMed links in the methods section to properly formatted references with numbers so these are listed in the reference list.

Answer:

Changes made.

Reviewer 1

  1. The findings presented in the supplementary files have not been described or referred to in the main text. Mention the findings and the significance of those experiments in the main text.

Answer:

Following Reviewer’s comment, we added a section (2.1) at the beginning of the Results, which makes a mention of Supplementary Fig. S1 and Supplementary Table S1. The mention of Supplementary Fig. S2 was made in the legend to Fig. 3. The other sections in the Results were renumbered.

Reviewer 1

  1. Present a time course of the ThT assay to show the development of β-sheet structures overtime (likely in a supplementary figure).

Answer:

As aggregation of titin is very fast, and first aggregates emerge already after 20–30 min, the fluorescence intensity maximum is observed in an hour. Then the intensity does not practically change for the subsequent 24 hours. A plot is attached in the "docx" file. 

Reviewer 1

  1. Make a distinction between ‘and’ and ‘or’ when two experimental conditions have been tested separately. This calls for ‘or’ instead of ‘and’. See tracked PDF.

Answer:

Slight changes were made; in particular, “and/or” were changed to “and”.

Reviewer 1

  1. Define mQ.

Answer:

“mQ” was changed to “deionized water (milli-Q)”.

Reviewer 1

  1. Discuss the limitations of the experimental design in your study. An obvious limitation is use of coverslips for growing the cells on two dimensions. Cells do not normally grow or function in two dimensions, considering the complex architecture of an organismal tissue. Discuss such limitations and their implications for the interpretation of the results. Discuss how the findings may be expanded or better described using a spheroid model of cell culture. This is important considering the new cell-culture models now used in the stem-cell field and cancer research, for example.

Answer:

Our results show that adhesion and spreading is impaired in cells of mesenchymal origin when they contact amyloid structures in a 2-D microenvironment. At the same time, in the organism, mesenchymal-origin cells are immersed into a 3-D extracellular matrix. Under these conditions, the inhibitory effect of amyloid structures on adhesion/spreading and, correspondingly, migration of cells can, presumably, be more pronounced than in a 2-D microenvironment. We believe that under 3-D conditions the signal of adhesion impairment should come from the entire surface of the cell, not from its basal part, as is the case in a 2-D microenvironment.

Discussing the obtained results, we note that suppression of cell adhesion using amyloid structures is of certain interest in the context of decreasing the migratory and invasive activity of tumour cells, which is increased as the result of the epithelial-mesenchymal transition. That is to say, amyloid structures in a tumour microenvironment can potentially decrease its invasive and metastatic potential.

We also added the following lines in the Discussion:

It should be noted that in the organism cells are in a 3-D microenvironment. Under these conditions, the inhibitory effect of amyloid structures on adhesion / spreading and, correspondingly, migration of cells can presumably be more pronounced than in a 2-D microenvironment – under 3-D conditions the impaired adhesion signal comes from the entire cell surface, not only from its basal part, as in 2-D.

Reviewer 1

  1. Discuss how do the RMS roughness values of 2.7–4.9 nm compare with an average cell dimensions grown in 2-D or 3-D culture models? Please discuss and provide information to make the context clear.

Answer:

The values of surface roughness we obtained are within the limits of 2.7–4.9 nm, while the size of cells in a 2-D model we described are about 100 µm. In Dang et al. (Dang, Y.; Quan, M.; Xing, C.M.; Wang, Y.B.; Gong, Y.K. Biocompatible and antifouling coating of cell membrane phosphorylcholine and mussel catechol modified multi-arm PEGs. J Mater Chem B. 2015, 3(11), 2350-2361. doi: 10.1039/c4tb02140a; we refer to this work in Discussion), the authors observe a similar anti-adhesive effect under analogous (2-D) conditions at the same roughness values as in our paper. Herewith, those authors use a polymer coating based on polyethylene glycol. As for studies on 3-D models, there are data of similar roughnesses in the literature. In particular, the authors of Chou et al. (Chou SF, Lai JY, Cho CH, Lee CH. Relationships between surface roughness/stiffness of chitosan coatings and fabrication of corneal keratocyte spheroids: Effect of degree of deacetylation. Colloids Surf B Biointerfaces. 2016 Jun 1;142:105-113. doi: 10.1016/j.colsurfb.2016.02.051) show on keratocyte spheroids that surface roughness plays a role in the formation of spheroids’ sizes. However, keratocytes are a convenient model for forming spheroids, unlike mesenchymal cells we use (e.g., smooth muscle cells or fibroblasts), which are unable of forming spheroids but form only various sizes of aggregates. It has been suggested in the literature that the roughness of the surface of a biopolymer coating has a significant effect on adsorption of fibronectin, which is a prerequisite for cell attachment and growth  (Velzenberger E, El Kirat K, Legeay G, Nagel MD, Pezron I. Characterization of biomaterials polar interactions in physiological conditions using liquid-liquid contact angle measurements: relation to fibronectin adsorption. Colloids Surf B Biointerfaces. 2009 Feb 1;68(2):238-44. doi: 10.1016/j.colsurfb.2008.10.022.).

This is how we answer this query – and our apologies if we did not completely seize the point of the matter.

Reviewer 1

  1. Discuss the implications of the surface roughness in thrombotic (fibrin) and atherosclerotic plaques? How do these compare with the model you used?

Answer:

When searching for answers to this query, we analyzed the following works:

- Burton HE, Cullinan R, Jiang K, Espino DM. Multiscale three-dimensional surface reconstruction and surface roughness of porcine left anterior descending coronary arteries. R Soc Open Sci. 2019 Sep 11;6(9):190915. doi: 10.1098/rsos.190915. PMID: 31598314; PMCID: PMC6774942.

- Owen DG, Schenkel T, Shepherd DET, Espino DM. Assessment of surface roughness and blood rheology on local coronary haemodynamics: a multi-scale computational fluid dynamics study. J R Soc Interface. 2020 Aug;17(169):20200327. doi: 10.1098/rsif.2020.0327. Epub 2020 Aug 12. PMID: 32781935; PMCID: PMC7482556.

- Timashev PS, Kotova SL, Belkova GV, Gubar'kova EV, Timofeeva LB, Gladkova ND, Solovieva AB. Atomic Force Microscopy Study of Atherosclerosis Progression in Arterial Walls. Microsc Microanal. 2016 Apr;22(2):311-25. doi: 10.1017/S1431927616000039. Epub 2016 Feb 4. PMID: 26843417.

Indeed, the surface roughness in vessels and arteries is an important parameter for diagnosing, in particular, atherosclerosis (Burton et al., 2019). The above works, using AFM and SEM, have shown that the vascular surface roughness values are within 200–1000 nm. The values of surface roughness obtained in our work are two orders of magnitude lower (2.7–4.9 nm). Thus, it is difficult to compare these data and it is difficult as yet to answer the question of the role of surface roughness in thrombotic and atherosclerotic plaques in cell adhesion.

Reviewer 1

  1. See tracked changes for examples of English-language changes in the text. I suggest seeking assistance from an academic English editor before resubmission.

Answer:

Unfortunately, we have not received the file with tracked changes. Still, we did our best to respond to Reviewer’s queries, including those of language and style.

Reviewer 2 Report

In the present manuscript, the authors assessed the in vitro effects of titin amyloid-like aggregates on RAOSMCs smooth muscle cells and human dermal fibroblast cell line. In particular, a cytotoxicity study has been conducted on smooth muscle cells in order to evaluate the direct role of SMT aggregates on cell viability. Based on these data, the morphology and the nanostructure of a formed SMT aggregate-coated surface was examined by atomic force microscopy, on which subsequently fibroblasts adhesion was finally evaluated. Moreover, confocal microscopy was used to observe every cellular morphological change.

The manuscript is well written and both results and discussion are well argued. In experimental terms, this work represents the continuation of the authors previous study and new information were added. Although other analyses should be performed to better characterize the obtained results, these represent a first step on the comprehension of amyloid fibrils pathogenic role. For this reason, in my opinion, the present paper should be published.

Minor revision:

In the discussion, the sentences of a specific paragraph related to cell attachment process:

“Interaction of cells with the surface is not straightforward. It occurs by means of proteins they secrete. The proteins, in turn, adsorb on the surface to form their own layer [54]. At the interaction of cells with the surface in vivo or in vitro first there occurs the adsorption of protein on the surface, which, in turn, provides related to cell attachment for cell adhesion, as well as contributes to the transfer of signals to the cell via cell adhesion receptors, mainly integrins”

should be revised. In order to facilitate readers, sentences and English form here could be changed.

In Figure 4, cell nuclei are not visible, as in Figure 7. Images could be changed or perfectioned.

Author Response

Reviewer 2

Minor revision:

In the discussion, the sentences of a specific paragraph related to cell attachment process:

“Interaction of cells with the surface is not straightforward. It occurs by means of proteins they secrete. The proteins, in turn, adsorb on the surface to form their own layer [54]. At the interaction of cells with the surface in vivo or in vitro first there occurs the adsorption of protein on the surface, which, in turn, provides related to cell attachment for cell adhesion, as well as contributes to the transfer of signals to the cell via cell adhesion receptors, mainly integrins”

should be revised. In order to facilitate readers, sentences and English form here could be changed.

Answer:

This text was changed as follows: “Cells interact with the surface via a layer of proteins, which they themselves secrete, e.g., cell fibronectin [55]. Their adhesion on the surfaces of these proteins is provided for by cell adhesion receptors, mainly integrins [38,56,57].”

Reviewer 2

In Figure 4, cell nuclei are not visible, as in Figure 7. Images could be changed or perfectioned.

Answer:

The images were replaced as recommended. Thanks!

Round 2

Reviewer 1 Report

I thank the authors for considering my suggestions. I apologise that the tracked PDF was not relayed to the authors. I reattached this file but this is of the previous version of the manuscript. The tracked changes are still helpful and relevant to the latest version of the manuscript. I did notice however that references 72 and 73 are still listed with the wrong format in the list of references. Perhaps the journal typesetting could capture and correct this mistake. Please seek assistance from an academic English editor before resubmission so that the text has been properly proofread and edited. 

Author Response

We have revised the manuscript according to reviewer comments. All changes in word-file and pdf-file are shown in red.

We only left unchanged the phrase: “No other mechanisms of smooth muscle cell death have been found [44]”, since these authors did not reveal other mechanisms of cell death.

We also ask the editors to check our English using editing service of the Journal.